# The Role of miR-375-3p, miR-210-3p and Let-7e-5p in the Pathological Response of Breast Cancer Patients to Neoadjuvant Therapy

**DOI:** 10.3390/medicina58101494

**Published:** 2022-10-20

**Authors:** Lorena Alexandra Lisencu, Andrei Roman, Simona Visan, Eduard-Alexandru Bonci, Andrei Pașca, Emilia Grigorescu, Elena Mustea, Andrei Cismaru, Alexandru Irimie, Cosmin Lisencu, Loredana Balacescu, Ovidiu Balacescu, Oana Tudoran

**Affiliations:** 1Department of Oncological Surgery and Gynecological Oncology, “Iuliu Hațieganu” University of Medicine and Pharmacy, 400012 Cluj-Napoca, Romania; 2Department of Radiology, The Oncology Institute “Prof. Dr. Ion Chiricuță”, 400015 Cluj-Napoca, Romania; 3Department of Genetics, Genomics and Experimental Pathology, The Oncology Institute “Prof. Dr. Ion Chiricuță”, 400015 Cluj-Napoca, Romania; 4Department of Surgical Oncology, The Oncology Institute “Prof. Dr. Ion Chiricuță”, 400015 Cluj-Napoca, Romania; 5Department of Pathological Anatomy, County Emergency Hospital, 400347 Cluj-Napoca, Romania; 6Research Center for Functional Genomics, Biomedicine and Translational Medicine, University of Medicine and Pharmacy “Iuliu Hatieganu”, 400037 Cluj-Napoca, Romania; 711th Department of Medical Oncology, University of Medicine and Pharmacy “Iuliu Hatieganu”, 400012 Cluj-Napoca, Romania

**Keywords:** breast cancer, neoadjuvant therapy, pathological complete response, miR-375-3p, let-7e-5p, MiR-210-3p

## Abstract

*Background and Objectives:* Prediction of response to therapy remains a continuing challenge in treating breast cancer, especially for identifying molecular tissue markers that best characterize resistant tumours. Microribonucleic acids (miRNA), known as master modulators of tumour phenotype, could be helpful candidates for predicting drug resistance. We aimed to assess the association of miR-375-3p, miR-210-3p and let-7e-5p in breast cancer tissues with pathological response to neoadjuvant therapy (NAT) and clinicopathological data. *Material and methods*: Sixty female patients diagnosed with invasive breast cancer at The Oncology Institute “Ion Chiricuță”, Cluj-Napoca, Romania (IOCN) were included in this study. Before patients received any treatment, fresh breast tissue biopsies were collected through core biopsy under echographic guidance and processed for total RNA extraction and miRNA quantification. The Cancer Genome Atlas Breast Invasive Carcinoma (TCGA-BRCA) database was used as an independent external validation cohort. *Results:* miR-375-3p expression was associated with more differentiated tumours, hormone receptor presence and lymphatic invasion. According to the Miller–Payne system, a higher miR-375-3p expression was calculated for patients that presented with intermediate versus (vs.) no pathological response. Higher miR-210-3p expression was associated with an improved response to NAT in both Miller–Payne and RCB evaluation systems. Several druggable mRNA targets were correlated with miR-375-3p and miR-210-3p expression, with upstream analysis using the IPA knowledge base revealing a list of possible chemical and biological targeting drugs. Regarding let-7e-5p, no significant association was noticed with any of the analysed clinicopathological data. *Conclusions:* Our results suggest that tumours with higher levels of miR-375-3p are more sensitive to neoadjuvant therapy compared to resistant tumours and that higher miR-210-3p expression in responsive tumours could indicate an excellent pathological response.

## 1. Introduction

Breast cancer (BC) is the most common cause of cancer among women worldwide, accounting for 15–20% of all cancer deaths in women [1]. Due to advances in diagnosis and treatment modalities, the prognosis of this disease has improved in recent years, with a 5-year survival rate of almost 90%. However, around 30% of breast cancer patients fail to respond to conventional treatments, leading to tumour progression [1].

Breast cancer is a highly heterogeneous disease; therefore, its treatment depends on multiple clinical and pathological factors such as tumour grade and hormone receptor (HR) status. Tumour characteristics and the extent of the disease direct the choice and timing of systemic treatments (chemotherapy, endocrine therapy or HER2-directed therapy). In the case of high-risk primary tumours or locally advanced breast cancer, neoadjuvant (preoperative) therapy is a frequently practical therapeutic approach as it offers the advantage of reducing the extent of surgery [2,3]. Furthermore, a tumour’s response to NAT can be used to guide adjuvant treatment selection and offer prognostic information regarding patient outcome [4].

The response to neoadjuvant therapy (NAT) is usually assessed clinically and pathologically. Pathological evaluation is the gold standard, as clinical evaluation can often misevaluate the response to NAT. While several systems have been proposed to evaluate the pathological response to NAT, the most used system for prognosis prediction are the Miller–Payne (MP) and the residual cancer burden (RCB) systems [5,6]. The MP system evaluates the changes in tumour cellularity between biopsy and surgery tissue. It has five grades as follows: 1 (no change), 2 (minor reduction in tumour cells, but ≤30%), 3 (reduction in tumour cells by 30–90%), 4 (reduction in tumour cells with >90%) and 5 (no detectable tumour cells). Grades 1–4 correspond to partial pathological response (pPR) while grade 5 means pathological complete response (pCR) [7,8]. The RCB system measures the primary tumour’s bidimensional size and cellularity and assesses lymph nodes’ involvement. The RCB index is classified as 0 (pCR), 1 (minimal residual disease), 2 (moderate residual disease) and 3 (extensive residual disease) [8,9]. The goal of NAT is pCR, as it plays an important prognostic role in BC patients. PCR is associated with improved overall survival and disease-free survival in comparison with those that do not achieve pCR and who have an unfavourable prognosis [3]. Due to its role in the prognosis of BC patients, predicting pCR is important in order to identify those patients that would benefit from NAT.

While routine HR, HER2 receptors, grading and Ki-67 assessment remain essential for treatment guidance, non-coding ribonucleic acids (RNA) have increased in popularity as their dysregulation has been associated with breast cancer pathogenesis. Microribonucleic acids (miRNA) are small, non-coding RNAs that regulate gene expression at the post-transcriptional level by binding to target messenger RNAs (mRNAs) and triggering their degradation. One of the first papers about the role of miRNA in cancer pathology demonstrated that miRNA is a better classifier than mRNA profiling when investigating poorly differentiated tumours, opening the way for using miRNA expression as a reliable marker for cancer diagnosis, prognosis and treatment response [10]. Since then, overwhelming data have indicated that miRNAs are involved in the regulation of processes such as proliferation, apoptosis and migration of cancer cells [3], having the potential of being oncogenic (oncomirs), tumour suppressors or both [1,11]. As key modulators of oncogenesis, miRNAs have been reported to have clinical utility in the diagnostic, prognostic and therapeutic approach of breast cancer patients [1,11], making them highly attractive as biomarkers for personalized medicine [1].

Several miRNAs have been reported to have predictive power in pathological response following NAT in breast cancer [12,13,14], with most of these studies being focused on neoadjuvant chemotherapy (NACT). However, recent treatment guidelines [15] encourage the administration of endocrine as well as targeted therapies concurrent with, or instead of, NACT to increase tumours’ sensitivity to treatment. Thus, there is an increasing need to further explore the role of these miRNAs as biomarkers of NAT response. Based on the existing literature, we have identified conflicting data regarding the prognostic role of several miRNAs. Of interest, miR-375-3p, miR-210-3p and let-7e have shown discrepancies regarding the clinical significance as prognostic biomarkers [16,17,18,19,20], being reported to have both increased and decreased expression associations with BC patients’ response to NAT. In this study, we aimed to assess the prognostic value of these highly controversial miRNAs, miR-375-3p, miR-210-3p and let-7e-5p in breast cancer tissues by investigating their expression association with patients’ pathological response to neoadjuvant therapy and clinicopathological features.

## 2. Materials and Methods

### 2.1. Breast Cancer Patients and Samples Collection

Sixty female patients diagnosed with invasive breast cancer at The Oncology Institute “Ion Chiricuță”, Cluj-Napoca, Romania (IOCN) were included in this study. The study was approved by the IOCN ethical committee (Approval No. 59/29.11.2016) and by the University of Medicine and Pharmacy Iuliu Hatieganu, Cluj-Napoca, Romania (Approval No. 290/09.09.2020). All patients were informed and gave their written consent for participation in the study following the Declaration of Helsinki. Before patients received any treatment, fresh breast tissue biopsies were collected through core biopsy under echographic guidance. The first core biopsy was sent for pathologic analysis, while a second biopsy was collected in RNAlater (Invitrogen, Thermo Fisher Scientific, Waltham, MA, USA) and stored in liquid nitrogen for transcriptomic studies.

### 2.2. RNA Extraction

Frozen biopsies were homogenized in TriReagent Solution (Ambion, Thermo Fisher Scientific, Waltham, MA, USA) using a Miccra D-1 (Miccra GmbH, Mullheim, Germany) polytron and processed for total RNA extraction using the classic phenol–chloroform method. The RNAs were quantified using NanoDrop ND-1000 (Thermo Scientific, Wilmington, DE, USA) and 2100 Bioanalyzer (Agilent Technologies, Santa Clara, CA, USA).

### 2.3. miRNA Expression Evaluation

Fifty nanograms (ng) of total RNAs were pre-amplified using universal RT miRNA primers to generate cDNAs following the TaqMan Advanced miRNA cDNA Synthesis Kit protocol (Thermo Fisher Scientific, Waltham, MA, USA). Next, 1:10 *v*/*v* diluted cDNAs and specific miRNA advanced assays were amplified with TaqMan Fast Advanced Master Mix (2X) (Thermo Fisher Scientific, Waltham, MA, USA) using the Light Cycler 480 device (Roche, Basel, Switzerland) with the following PCR settings: 55 °C for 2 min to remove RNA contaminants; 95 °C for 20 s for Taq polymerase amplification; and 40 cycles of 95 °C for 3 s followed by 60 °C for 30 s for PCR amplification. The ∆∆Ct method was used for miRNA relative quantification by reporting the Ct values of the miRNAs of interest to miR-16-5p Ct values.

### 2.4. TCGA Data Analysis

The Cancer Genome Atlas Breast Invasive Carcinoma (TCGA-BRCA) expression data (miRNA and mRNA) and their clinical information were obtained from National Cancer Institute Genomic Data Commons (NCI GDC) data portal (https://portal.gdc.cancer.gov/, accessed on 19 April 2019) and cBioPortal for Cancer Genomics (https://www.cbioportal.org/, accessed on 19 April 2019). The miRNA-seq data, expressed as reads per million and fragments per kilobase millions mRNA-seq data, were filtered and log2(x + 1)-transformed. After processing, a miRNA dataset containing 916 tumoral samples and 93 standard samples and an mRNA dataset of 983 tumoral samples were retained for subsequent analysis. Pearson correlation was used to test potential miRNAs–mRNA associations and intersected with validated miRNA–target interactions retrieved from the miRTarBase. The Ingenuity Pathway Analysis (IPA, Qiagen, Redwood City, CA, USA) upstream analysis module was used to interrogate for possible targeting drugs.

### 2.5. Statistical Analysis

The correlation between clinicopathological characteristics and tissue miRNAs expression was evaluated with the Mann–Whitney U test for two categorical variables or the Kruskal–Wallis test. It was followed by Dunn’s multiple comparison post hoc test in the case of three or more categorical variables based on the data distribution. A *p*-value less than 0.05 was considered statistically significant. Fold regulation (FR) was calculated as the ratio between mean value of the interest group and the reference group.

## 3. Results

### 3.1. Patient and Tumour Characteristics

The clinicopathological features of the 60 included patients are summarized in Table 1. The median age of the patients was 60 (29–77), with most of the patients (73.33%) being over 50 years old at the time of diagnosis. Over 85% of the patients had moderately to poorly differentiated carcinomas, and 61.67% presented Ki-67 higher than 20. Most of the patients had luminal tumours (76.67%), with luminal B being the predominant subtype (46.67%). Over 85% of the patients were already in advanced clinical stages at diagnosis (>II). Of the 60 investigated patients, 49 received NAT: 32 received chemotherapy alone, 7 received hormonal therapy, 4 also received Her2 targeted therapy, while 5 received combinations of regimens. TNM staging was retained for prognostic information (primary and post-NAT surgery), while the Miller–Payne and RCB systems were used to evaluate the pathological response of the patients to NAT. According to the Miller–Payne evaluation, 14 patients did not respond to NAT, 4 presented a minor response, 13 had an intermediate response, and 13 had almost complete pathological response. According to the RCB classification system, 13 patients reached a high pathological response, 16 were therapy-resistant, and 15 had a partial response.

### 3.2. Investigation of miRNA Expression in Breast Cancer Specimens

The association between tissue miRNA expression and the clinicopathological data of the patients included is presented in Table 2. No correlations were observed between the investigated miRNA expression and the age of the patients at a 50-years-of-age cut-off value. Additionally, no significant clinicopathological associations were found for let-7e-5p expression. Lower miR-375-3p expressions were associated with higher tumour grading and KI67 proliferation index. ER- and PR-positive tumours had a higher miR-375-3p expression, with significantly decreased expression in TNBC compared to luminal subtypes. Except for miR-375-3p expression with lymphatic positivity, no other significant correlations were demonstrated between tissue miRNAs expression and TNM staging, neither for clinical nor pathological evaluations. According to the Miller–Payne system, patients with a high pathological response to NAT had lower miR-375-3p and higher miR-210-3p expressions compared to intermediate- and low-responding patients. Higher miR-210-3p expression was observed for high responders versus partial responders in the RCB evaluation system.

Analysis of the miRNAs expression in the TCGA database was performed as an independent external validation cohort. Increased miRNA expression was observed between tumour and standard samples (Appendix A), with luminal tumours having significantly higher miR-375-3p and lower miR-210-3p expression (Figure 1, Appendix A) compared to the basal-like subtype. No significant miRNA expression differences (FR cut-off > 1.5) were observed for patients with positive lymph nodes (Appendix A) or metastatic diseases (Appendix A). Patients with advanced diseases had higher miR-210-3p expression (Figure 2, Appendix A), while a slight decrease in miR-210-3p expression was associated with a better survival (Figure 3, Appendix A).

In order to explore the possible mechanisms mediated by the investigated miRNAs, a correlation analysis between miRNA and mRNA expression in the TCGA cohort was undertaken. The significantly correlated genes were intersected with the validated mRNA targets downloaded from the miRTarBase. Only the validated target genes that were inversely correlated with miRNA expression and with a correlation coefficient under −0.3 were considered of interest (Appendix A). A total of 17 possible mRNA targets for miR-375-3p and 9 for miR-210-3p were identified (Figure 4A). No significantly correlated genes were observed for let-7e-5p. The number of identified genes was too small to run a GSEA analysis; therefore, no specific molecular mechanisms could be attributed to either miRNA. Upstream analysis using the IPA knowledge base identified a list of 127 possible chemical and biological targeting drugs (Figure 4B).

## 4. Discussion

Different miRNA signatures have been associated with response to chemo-endocrine or radiotherapy in breast cancer [11], emerging as valuable biomarkers for a personalized therapeutic approach of the disease [1,23]. This study reports the expression profiles of three miRNAs involved in the treatment response of BC patients. The analysis explored the association of miRNA expression with the pathological response to NAT and their correlation with the clinicopathological data of the patients.

The patients’ pathological response to NAT was assessed using both MP and RCB systems. However, both of them have limitations. The MP system ignores the involvement of the axillary lymph nodes; therefore, the prognosis can be overrated in lymph-node-positive patients [8]. The RCB classification showed a better performance than the MP system, especially for the TNBC subtype [8], but it is limited to anatomical factors without considering the biological ones [8]. To improve its value, assessment of Ki-67 expression after treatment in combination with the RCB index might improve the prediction of survival outcomes [24]. Of the three investigated miRNAs, miR-375-3p and miR-210-3p were significantly associated with MP response, while miR-210-3p expression was also significantly associated with RCB.

MiR-375-3p is dysregulated in various types of cancer. It is involved in epithelial-to-mesenchymal transition (EMT), and is associated with increased invasiveness potential while also being correlated with refractory response to chemotherapy [25]. MiR-375 is a known tumour suppressor. In hepatocellular carcinoma (HCC), it inhibits the autophagy and tumour growth. Moreover, miR-375 promotes the release of mitochondrial apoptotic proteins, reducing the viability of HCC cells in hypoxic conditions. In HCC, miR-375 was downregulated. Autophagy is an adaptive mechanism of the tumour cells that helps them to survive in the tumour microenvironment conditions by reducing apoptosis and enhancing the elimination of the injured mitochondria [26]. Although miR-375 could be related to treatment response by mitochondria reprogramming, to date, there are no data presenting evidence about the role of mir-375 in inducing NAT response through mitochondria reprogramming in breast cancer. Despite the well-documented role as a tumour suppressor, in breast cancer, miR-375-3p is upregulated [16,27,28] and is highly expressed in hormone-receptor-positive breast tumours [29] and lymph-node-positive patients [29]. The present results are in line with these findings. The upregulated expression of miR-375-3p in breast cancer suggests a potential oncogenic activity [16]. Predicted miR-375-3p targets were downloaded from the miRTarBase and intersected with inversely correlated miRNA-mRNA genes from TCGA. Using GSEA, we interrogated Reactome, Kyoto Encyclopedia of Genes and Genomes (KEEG), and Gene Ontology (GO) databases to explore gene functionality. However, the gene list was too small to generate any significant signalling pathways associated with the identified genes. Thus, the regulatory role of miR-375-3p in breast cancer remains unclear.

Multiple miR-375-3p-mediated therapy resistance mechanisms have been described. Generally, miR-375-3p expression is downregulated in drug-resistant BC cells, while its overexpression has been shown to increase cells’ sensitivity to chemo-endocrine or targeted therapy. Mir-375-3p-mediated targeting of YBX1 [30] and JAK2 genes [31] has led to increased sensitivity to Adriamycin and paclitaxel first-line treatments. In fulvestrant-resistant BC cells, the overexpression of miR-375-3p inhibited cell growth and autophagy by silencing autophagy-related proteins [30]. Furthermore, by targeting HOXB3 (17) or MDTH gene expression [32], miR-375-3p has decreased EMT, stem features and resistance to tamoxifen in ER-positive BC cells. Moreover, epigenetic silencing of miR-375-3p induced trastuzumab resistance in HER2-positive BC by targeting IGF1R [33]. All this evidence suggests that miR-375-3p might serve as a potential therapeutic approach for the treatment of resistant breast cancer and as a prognostic marker of therapy.

According to the MP evaluation system, in the present cohort, higher miR-375-3p expression was calculated for patients with intermediate response to NAT compared to nonresponders or good responders. Consistent with previous reports, lower miR-375-3p expression levels were observed in the resistant-to-NAT tumour group, while higher expression levels were associated with an improved response. Notably, low levels of miR-375 were also observed in the high-responsive group of patients.

The existing literature regarding the prognostic potential of miR-375-3p is controversial. Furthermore, most reports are based on circulating miR-375-3p levels. According to a three-year follow-up study, patients with relatively higher tumour miR-375-3p expression had a worse survival rate and less survival time, namely, a worse prognosis [34]. On the other hand, a lower expression of circulating miR-375-3p was associated with incomplete response to NAT, while increased expression was noticed in patients achieving pCR after NAT [35]. Similarly, Wu et al. [36] reported that miR-375-3p prevalence in circulation was associated with better clinical outcomes, complete response to NAC, and an absence of relapse. Simultaneously, lower levels of miR-375-3p were noted in therapy-resistant HER2-positive patients. In nonresponder luminal B HER2- patients, NAT can induce the upregulation of circulating miR-375-3p, and this change might be associated with a good response to NAT [35].

Moreover, miR-375-3p association with NAT response seems to be subtype-specific. A lower expression of miR-375-3p was correlated with an increased risk of disease relapse in luminal B patients, while in luminal A, patients with lower miR-375-3p expression were found to be more sensitive to NAC [37]. However, when comparing circulating miRNA levels with tumour levels, it should be considered that the cellular source of circulating miR-375-3p remains unknown; their prevalence may not reflect expression in the primary tumour but rather a combination with other cell types, such as immune cells.

High expression of miR-210-3p in human cancers has become a predictive marker of tumour hypoxia, increasing experimental evidence supporting its clinical relevance. In BC, miR-210-3p is overexpressed in tumour tissues, specifically in triple-negative and HER2+ tumours compared [38], while miR-210-3p upregulation was associated with drug-resistant breast cancer cells [18,39].

The literature data regarding the association of miR-210-3p expression and patient response to neoadjuvant treatment are conflicting, highlighting that both increased and decreased miR-210-3p expression have shown discrepancies regarding the clinical significance as prognostic biomarkers. In HER2-positive BC patients, higher circulating miR-210-3p levels were noticed before surgical excision in patients with residual disease and lymph node metastasis [18]. Muller et al. reported increased miR-210-3p circulating levels following NAT. However, no association between miR-210-3p levels and pCR was observed [40]. Conversely, higher circulating miR-210-3p levels have been associated with residual disease following trastuzumab-combined NAC in BC patients [18,41]. Similar results were also described in ER-positive BC patients treated with tamoxifen; miR-210-3p expression was linked with poor clinical outcomes and an increased risk of relapse [42]. A meta-analysis revealed that miR-210-3p overexpression correlated with poorer survival in TNBC patients and associated circulating miR-210 expression with resistance to doxorubicin, cyclophosphamide, cisplatin and paclitaxel [43]. When quantified in tissue samples, no clinical association with NAT response was observed in miR-210-3p expression levels in BC formalin-fixed paraffin-embedded tissues [40]. However, miR-210-3p expression was suggested to represent a marker for predicting metastasis development and a worse prognosis in patients treated with taxanes in adjuvant settings [44]. Bioinformatic analysis identified microtubule regulation, drug efflux pathways and NRF2-mediated oxidative stress response as the most significant associated pathways between miR-210-3p signalling and docetaxel resistance [44].

Analysis of the miR-210-3p expression in the present cohort was performed to further elucidate the controversy surrounding this miR’s ability to predict cPCR. Patients with almost complete or partial pathological responses had significantly higher miR-210-3p expression levels than nonresponders. MiR-210-3p’s role as an oncogene is well-characterized; however, it was also suggested to act as a tumour suppressor [19]. For example, in oesophageal squamous cell carcinoma, miR-210-3p has been shown to inhibit cancer cell survival and proliferation by inducing cell death and cell cycle arrest in G(1)/G(0) and G(2)/M through FGFRL1 downregulation [19]. More recently, Bar I et al. [45] showed that in TNBC patients, miR-210-3p is expressed by both tumour cells and the tumour microenvironment (TME) that is more likely to be regulated by a mechanism independent of HIF-1 alpha. MiR-210-3p has multiple functions, including the regulation of the immune response and increasing data support the concept that immunologically “hot” tumours are more responsive to chemotherapy [46].

Let-7e-5p is one of the first discovered miRNAs [47]. This is a tumour suppressor gene that targets essential pathways involved in tumorigeneses such as Janus protein tyrosine kinase (JAK), c-Myc and signal traducer and activator of transcription 3 (STAT3). The literature reports about let-7e-5p expression during NAT are scarce. In patients with a lower Ki-67 and pCR, a decrease in let-7e-5p expression was noticed after NAT [48]. Lv. J. et al. showed that let-7e-5p expression is down-regulated in chemoresistant tumours, while decreased expression was associated with a worse prognosis [20]. The present analysis did not find any significant association between let-7e-5p expression and pathological response to NAT or with the clinicopathological features of the patients, neither in the given cohort nor in the TCGA database.

Our study is limited by the patients’ heterogeneity and the relatively small sample size cohort and, thus, the present analysis has low statistical power. However, to the best of our knowledge, this is the first study that illustrates a putative relationship between miR-375-3p and miR-210-3p and breast tumours’ pathological response to neoadjuvant therapy. Increasing the number of patients would allow for more homogeneous groups and subsequent differential analysis based on the administered type of therapy. Of great interest, prognostic markers for patients’ response to combined regimens are largely unexplored.

## 5. Conclusions

Based on documented mechanistic actions of the two miRNAs, our results suggest that tumours with higher levels of miR-375-3p are more sensitive to neoadjuvant therapy compared to resistant tumours, and that higher miR-210-3p expression in responsive tumours could indicate immunologically “hot” tumours. These findings suggest the potential role of these two miRNAs in stratifying BC patients that will respond to NAT. However, as data regarding these two miRNAs are controversial, further studies are needed to elucidate their complex role in mediating BC patients’ response to neoadjuvant therapy.

## Figures and Tables

**Figure 1 medicina-58-01494-f001:**
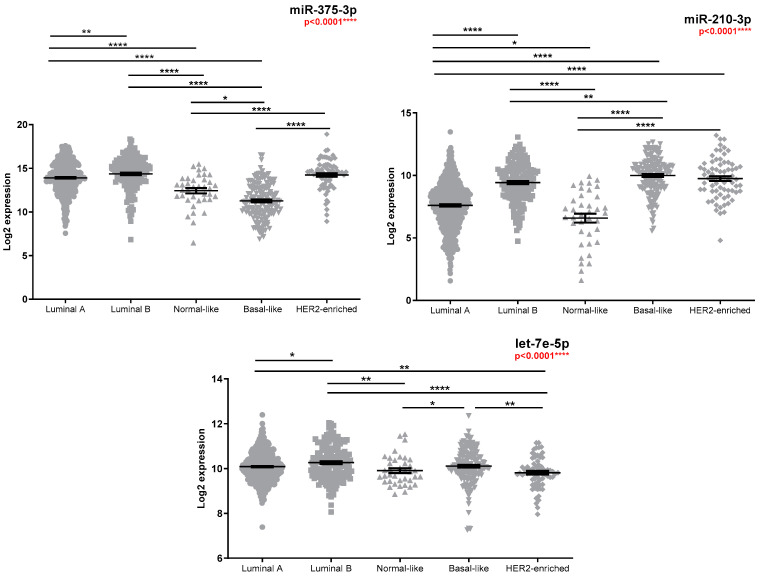
TCGA miR-375-3p, miR-210-3p, and let-7e-5p expression according to PAM50 classification (n = 470 luminal A, n = 162 luminal B, n = 38 normal-like, n = 159 basal-like, and n = 71 HER2-enriched). Kruskal–Wallis test was used for between-groups comparison and Dunn test for multiple comparisons (* *p* < 0.05, ** *p* < 0.01, **** *p* < 0.0001). FR values between groups of interest along with individual *p*-values obtained by Dunn test are reported in Appendix A.

**Figure 2 medicina-58-01494-f002:**
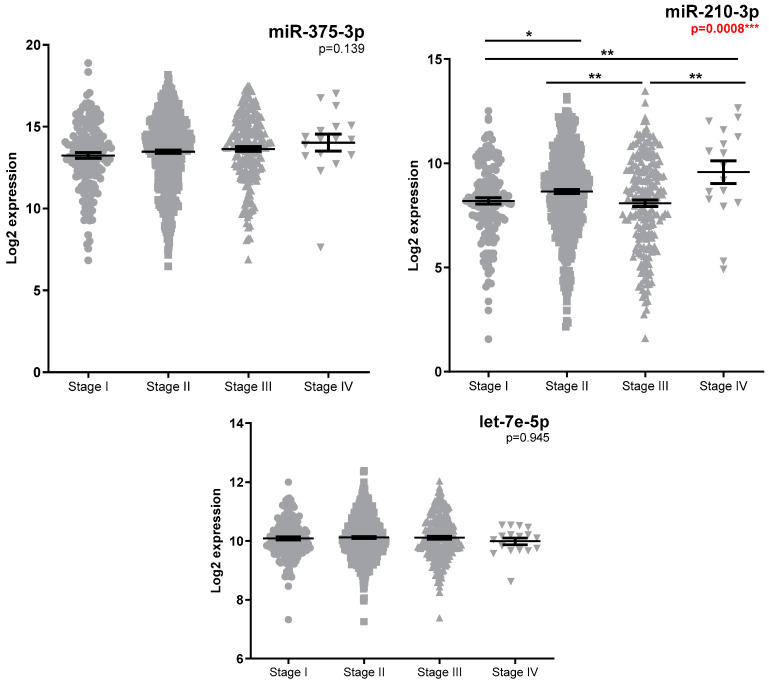
TCGA miR-375-3p, miR-210-3p, and let-7e-5p expression according to pathologic tumour stage (n = 158 stage I, n = 510 stage II, n = 210 stage III, and n = 17 stage IV). Kruskal–Wallis test was used for between-groups comparison and Dunn test for multiple comparisons (* *p* < 0.05, ** *p* < 0.01, *** *p* < 0.001). FR values between groups of interest along with individual *p*-values obtained by Dunn test are reported in Appendix A.

**Figure 3 medicina-58-01494-f003:**
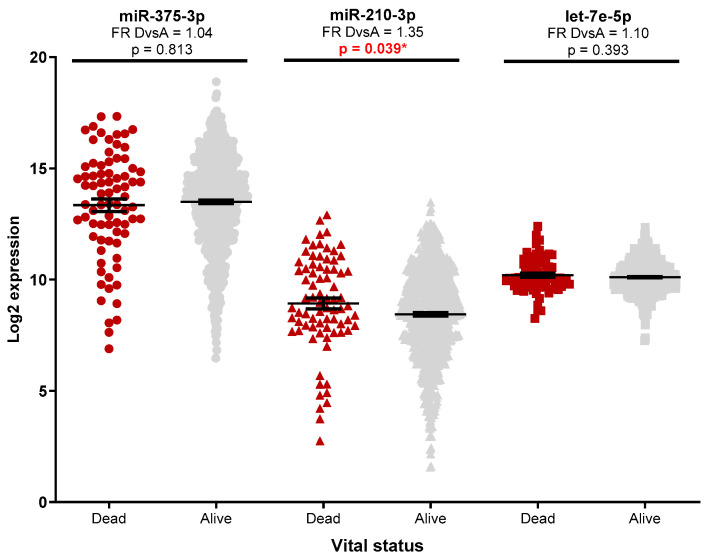
TCGA miR-375-3p, miR-210-3p, and let-7e-5p expression according to vital status. Groups: D—dead, with tumour or with a new tumour event (n = 77); A—alive, tumour-free, without a new event tumour (n = 724). Mann–Whitney test was used for group comparison (* *p* < 0.05).

**Figure 4 medicina-58-01494-f004:**
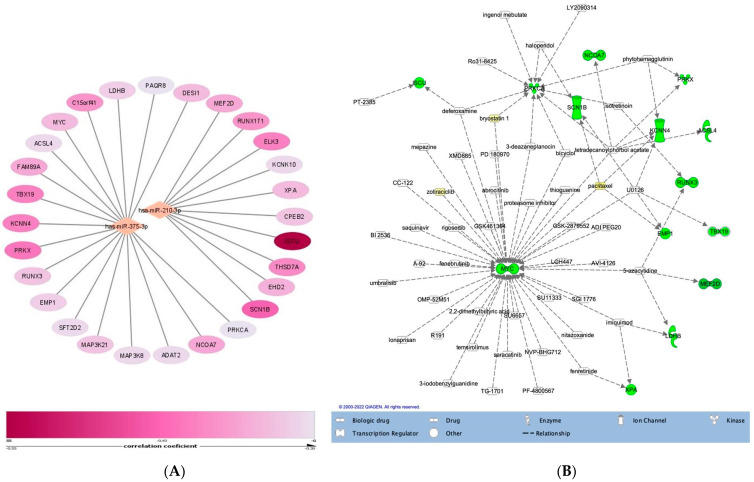
(**A**) miR-375-3p- and miR-210-3p-correlated validated mRNA targets in the TCGA cohort. Node fill colour was continuously mapped according to the correlation coefficient, (**B**) Drugs that target mRNA downstream genes.

**Table 1 medicina-58-01494-t001:** Clinicopathological data of the patients included in the study.

Variable	Patients Characteristics
n = 60	
Age	
≤50	16 (26.67%)
>50	44 (73.33%)
Grading	
G1	7 (11.67 %)
G2	32 (53.33%)
G3	20 (33.33%)
NA	1 (1.67%)
Ki-67	
≤20	22 (36.67%)
>20	37 (61.67%)
NA	1(1.67%)
Molecular Subtype	
Luminal A	18 (30%)
Luminal B	28 (46.67%)
HER2+	3 (5%)
TNBC ^1^	8 (13.33%)
NA ^2^	3(5%)
Tumour size (c ^3^)	
cT1	6 (10%)
cT2	29 (8.33%)
cT3	8 (13.33%)
cT4	11 (18.33%)
NA	6 (10%)
Lymph nodes (c)	
cN0	15 (25%)
cN1	15 (25%)
cN2	21 (35%)
cN3	3 (5%)
NA	6 (10%)
Metastasis (c)	
cM0	49 (81.67%)
cM1	1 (1.67%)
NA	10 (16.67%)
Clinical stage	
Stage I	5 (8.33%)
Stage II	19 (31.67%)
Stage III	27 (45%)
Stage IV	1 (1.67%)
NA	8 (13.33%)
Tumour size (p ^4^)	
pT0	7 (11.67%)
pT1	23 (38.33%)
pT2	18 (30%)
pT3	1 (1.67)
NA	11 (18.33%)
Lymph nodes (p)	
pN0	26 (43.33%)
pN1	13 (21.67%)
pN2	8 (13.33%)
pN3	4 (6.67%)
NA	9 (15%)
Lymphatic invasion (p)	
L0	30 (50%)
L1	21 (35%)
NA	9 (15%)
n = 49	
Neoadjuvant therapy	
Only CT ^5^	32 (65.31%)
Only ET ^6^	7 (14.29%)
CT + HT	3 (6.12%)
Combinatory CT/ET/RTE ^7^	2 (4.08%)
Her2+ TT ^8^	4 (8.16%)
NA	1 (2.04%)
Miller–Payne system	
Grade 1	14 (28.57%)
Grade 2	4 (8.16%)
Grade 3	13 (26.53%)
Grade 4	5 (10.2%)
Grade 5	8 (16.33%)
NA	5 (10.2%)
RCB ^9^	
RCB 0	8 (16.33%)
RCB-I	5 (10.2%)
RCB-II	15 (30.61%)
RCB-III	16 (32.65%)
NA	5 (10.2%)

^1^ TNBC—triple negative breast cancer, ^2^ NA—nonassessable, ^3^ c—clinic, ^4^ p—pathologic, ^5^ CT—chemotherapy, ^6^ HT—hormonal therapy, ^7^ RTE—external radiotherapy, ^8^ TT—targeted therapy, ^9^ RCB—residual cancer burden.

**Table 2 medicina-58-01494-t002:** Tissue miRNA expression in relation to the clinicopathological data.

Clinicopathological Features	miR-375-3p	miR-210-3p	let-7e-5p
	*p*-Value	*p*-Value	*p*-Value
Age ≤50 vs. >50	0.116	0.709	0.682
Grading_Biopsy G1 vs. ^1^ G2 vs. G3	**0.007 **** **FR ^2^ G3 vs. G1 * = −2.42** **FR G3 vs. G2 ** = −3.08**	0.783	0.118
ER ^3^ [21] Negative (<1) vs. Positive (≥1)	**0.0008 ***** **FR ER+ vs. ER- = 6.13**	0.970	0.209
PgR ^4^ [21] Negative (<1) vs. Positive (≥1)	**0.009 **** **FR PgR+ vs. PgR- = 2.28**	0.384	0.305
Ki-67 [22] Negative (≤20) vs. Positive (>20)	**0.042 *** **FR Ki-67+ vs. Ki-67- = −1.49**	0.187	0.052
Molecular Subtype LuminalA (LumA) vs. LuminalB (LumB) vs. TNBC ^5^	**0.004 **** **FR TNBC vs. LumA *** = −9.80** **FR TNBC vs. LumB ** = −7.28**	0.332	0.296
cT ^6^ T1 vs. T2 vs. T3 vs. T4	0.341	0.278	0.885
cN ^7^ Negative (N0) vs. Positive (N1 + N2 + N3)	0.799	0.879	0.350
Clinical Stage Early (Stage I + II) vs. Advanced (Stage III + Stage IV)	0.343	0.834	0.463
pT ^8^ T0 vs. T1 vs. T2	0.261	0.555	0.749
pN ^9^ Negative (N0) vs. Positive (N1 + N2 + N3)	0.469	0.815	0.929
Lymphatic Invasion L0 vs. L1	**0.043 *** **FR L1 vs. L0 = 1.94**	0.064	0.853
Miller–Payne Low (1+2) vs. Intermediate (3) vs. High (4 + 5)	**0.019 *** **FR High vs. Intermediate * = −3.03** **FR Intermediate vs. Low * = 3.14**	**0.016 *** **FR High vs. Low * = 2.54** **FR Intermediate vs. Low * = 2.06**	0.409
RCB ^10^ 0/I vs. II vs. III	0.416	**0.023 *** **FR 0/I vs. III ** = 2.87**	0.475

^1^ vs.—versus, ^2^ FR—fold regulation, ^3^ ER—oestrogen receptor, ^4^ PgR—progesterone receptor, ^5^ TNBC—triple negative, ^6^ cT—clinic tumour, ^7^ cN—clinic lymph node, ^8^ pT—pathologic tumour, ^9^ pN—pathologic lymph node, ^10^ RCB—residual cancer burden. * The table shows the *p*-values for all the compared groups, and where these differences were statistically significant, the expression level of that miRNA (FR) in the interest group versus the reference group was calculated. According to data distributions, differences in expression in the case of two groups were evaluated with the Mann–Whitney test and for three groups with the Kruskal–Wallis test, followed by Dunn’s multiple comparison post hoc test. In the case of three or more groups, the asterisk next to the FR value is associated with the *p*-value obtained by the Dunn test (* *p* < 0.05, ** *p* < 0.01, *** *p* < 0.001).

## Data Availability

Not applicable.

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
