# Peer review of "The Role of miR-375-3p, miR-210-3p and Let-7e-5p in the Pathological Response of Breast Cancer Patients to Neoadjuvant Therapy"

_medicina, 2022, doi:10.3390/medicina58101494_

Round 1
Reviewer 1 Report
In this manuscript, the authors aimed at assessing the prognostic value of three highly controversial miRNAs, miR-375-3P, miR-210-3p and let-7e-5p in invasive breast cancer. Using 60 patient samples and TCGA database as an independent validation cohort, the authors investigated the expression profiles of the three miRNAs and their association with the patients' pathological response to neoadjuvant therapy and clinico-pathological features. The authors concluded that tumors that are more sensitive to neoadjuvant (NAT) therapy have higher level of miR-375-3p than the resistant tumors and tumors with higher miR-210-3p expression could have a better response to NAT at the pathological level. Overall, this is a well-designed study and a well-written manuscript. The analyses are thorough and the discussion has comprehensively talked about the limitation of the study and the conflicts with other published studies. This reviewer only has two recommendations which would hope to make some improvements to the manuscript
1. The authors may consider adding some background knowledge of the three miRNAs in the "Introduction" part of the manuscript so the readers wouldn't be wondering why these three miRNAs are specifically investigated here
2. Are the criteria to classify the tumors as stage I....stage IV in Figure 2 same as those to stage tumors as S1...S4 in Table 2? If so, the authors may consider using the same format to present that data so that they can be comparable. For example, stage I and stage II in figure 2 can be combined to early stage and stage III and stage IV to advanced stage or separate early and advanced stages in table 2 into four stages for comparison.
Author Response
We would like to thank you for your time spent on reviewing our manuscript! Your comments and suggestions helped us to improve our paper!
In this manuscript, the authors aimed at assessing the prognostic value of three highly controversial miRNAs, miR-375-3P, miR-210-3p and let-7e-5p in invasive breast cancer. Using 60 patient samples and TCGA database as an independent validation cohort, the authors investigated the expression profiles of the three miRNAs and their association with the patients' pathological response to neoadjuvant therapy and clinico-pathological features. The authors concluded that tumors that are more sensitive to neoadjuvant (NAT) therapy have higher level of miR-375-3p than the resistant tumors and tumors with higher miR-210-3p expression could have a better response to NAT at the pathological level. Overall, this is a well-designed study and a well-written manuscript. The analyses are thorough and the discussion has comprehensively talked about the limitation of the study and the conflicts with other published studies. This reviewer only has two recommendations which would hope to make some improvements to the manuscript
- The authors may consider adding some background knowledge of the three miRNAs in the "Introduction" part of the manuscript so the readers wouldn't be wondering why these three miRNAs are specifically investigated here
We have restructured the Introduction according to both reviewers indications. We have highlighted the controversial role of the three miRNAs based on literature evidence, which determined their further investigation in the present study. Further background information is described for each miRNA in the Discussion section.
- Are the criteria to classify the tumors as stage I....stage IV in Figure 2 same as those to stage tumors as S1...S4 in Table 2? If so, the authors may consider using the same format to present that data so that they can be comparable. For example, stage I and stage II in figure 2 can be combined to early stage and stage III and stage IV to advanced stage or separate early and advanced stages in table 2 into four stages for comparison.
We have modified clinical stage nomenclature to Stage I…IV throughout the manuscript (Tables 1 and 2). We have combined the clinical stages in early and advanced in our cohort due to the small number of cases in each category, which have made statistical comparison redundant. As the TCGA database has sufficient number of cases in each stage for statistical comparison, we think that individual stages comparisons are more relevant than combining them as we did for our cohort.
Reviewer 2 Report
Summary: A retrospective analysis to detect the role of three miRNAs (miRs) in defining breast cancer characteristics and response to the Neoadjuvant chemotherapy. The study showed that miR-375 overexpression is associated with lower grade tumors, higher hormone receptor expression, and less proliferation. Also, it demonstrated that higher miR-375 and 210 expression are associated with better response to the Neoadjuvant therapy.
General evaluation: A valuable research in terms of methodology and clinical implications.
Comments:
1. Main title: please designate the type of neoadjuvant therapy. It might be confusing for the readers and provide a question that, Is it neoadjuvant chemotherapy or neoadjuvant radiotherapy?
2. The abstract doesn't give any information on the factor let-7e-5p, which has been mentioned in the main title.
3. The study found that "miR-375-3p expression was associated with more differentiated tumors and hormone receptors presence". Its is also found that "higher levels of miR-375-3p are more sensitive to neoadjuvant therapy". This finding doesn't make sense. According to the literature it is established that patients with hormone recoptor presnce have poor response to the neoadjuvant chemotherapy (PMID: 30255695). How the authors can justify this finding? The authors can refer to the comment (12) for discuss on this finding in the Discussion section.
4. The Introduction section contains much unnecessary contents. Please summarize the Introduction and provide information around the main title of the study. The authors can organzie the Introduction section as follows:
- The importance of breast cancer in nowadays living
- The rate of locally advanced breast cancer (LABC)
- the importance of neoadjuvant chemotherapy in LABC
- grading systems of response to NAC.
- the importance of predicting the pathologic response to neoadjuvant chemotherapy (NAC)
- the lack of information regarding the the pathologic response to NAC
- Short definition of miRNAs.
- The current knowledge on the role of miRNAs in predicting response to chemotherapy (if no study on breast cancer, authors can use the other cancers in this section)
- the aim of the study
5. Introduction: "Treatment options for BC are represented by: surgery, radiotherapy, hormone or tar- 78 geted therapy and chemotherapy" Targeted therapy is not a common option in patients undergoing neoadjuvant chemotherapy.
6. Introduction: The rationale behind selecting the miRNA 375, 210, and let-7e in this study is not explained.
7. Table 1. Please add the full term of TT in the footnote.
8. Table 1 footnote: please correct the following typographical error:
"TNBC- Triple negativ" => negative
9. Table 2. Please choose a summarized heading (one line), and add the description to the footnote.
10. Table 2. Please add the reference for defining ER positivity >10% and PR positivity >20%. Based on my knowledge, ER and PR positivity are defined when >1% of the cancer cell are positive. This comment can be extended to KI-67. According to the literature, KI-67 >13% is considered high, not 20%.
11. Discussion: Please add more details regarding the mechanistic pathways involved in the cancer biology. In case of lack of data, the authors can extrapolate the condition from other malignancies.
12. Discussion: In addition, it is recommended the authors add their own opinions for the machisms of action of the evaluated miRs in cancer biology. For example, recent literature have shown that mitochondria is at the center of cancer biology, in order that enhanced mitochondria biology can reduce cancer differentiation and improves treatment resistance ( https://www.preprints.org/manuscript/202201.0171/v3. and https://pubmed.ncbi.nlm.nih.gov/34890822/). On the other hand, it has been shown that miR-375 has an inhibitory effect of mitochondria biogenesis (https://pubmed.ncbi.nlm.nih.gov/22504094/). Therefore, this interaction can justify how miR-375 overexpression is associated with lower grade breast cancers and also high hormone receptor expression. Utmost, this interaction can justify how miR-375 can improve the response to neaodjuvant chemotherapy (https://www.preprints.org/manuscript/202201.0171/v3).
13. Discussion: Please add the study strengths and limitations and provide suggestions for future works.
Author Response
We would like to thank you for your time spent on reviewing our manuscript! Your comments and suggestions helped us to improve our paper!
Summary: A retrospective analysis to detect the role of three miRNAs (miRs) in defining breast cancer characteristics and response to the Neoadjuvant chemotherapy. The study showed that miR-375 overexpression is associated with lower grade tumors, higher hormone receptor expression, and less proliferation. Also, it demonstrated that higher miR-375 and 210 expression are associated with better response to the Neoadjuvant therapy.
General evaluation: A valuable research in terms of methodology and clinical implications.
Comments:
1. Main title: please designate the type of neoadjuvant therapy. It might be confusing for the readers and provide a question that, Is it neoadjuvant chemotherapy or neoadjuvant radiotherapy?
While we agree that the readers might be confused before reading the manuscript, we have explained in the Lines 316-317 (Of the 60 investigated patients, 49 received NAT: 32 received chemotherapy alone, 7 received hormonal therapy, 4 also received Her2 targeted therapy, while 5 received combinations of regimens) that the patients received multiple types of therapies. While it would be interesting to run a differential analysis based on the administered type of therapy, the number of patients in each category is not sufficient for statistical analysis. This aspect has been discussed in the manuscript as a future direction. Furthermore, we highlighted in the introduction that the current treatment guidelines include combined therapeutic regimens to further enhance chemosensitivity, thus the need to assess the prognostic role to neoadjuvant endocrine therapy, as well as targeted therapies and combined regimens as NAT.
The abstract doesn't give any information on the factor let-7e-5p, which has been mentioned in the main title.
We have added a phrase in lines 44-45 that states that no significant association was noticed between let-7e-5p and the clinico-pathological data.
The study found that "miR-375-3p expression was associated with more differentiated tumors and hormone receptors presence". Its is also found that "higher levels of miR-375-3p are more sensitive to neoadjuvant therapy". This finding doesn't make sense. According to the literature it is established that patients with hormone recoptor presnce have poor response to the neoadjuvant chemotherapy (PMID: 30255695). How the authors can justify this finding? The authors can refer to the comment (12) for discuss on this finding in the Discussion section.
The indicated study refers strictly to the response of hormone receptor positive patients to neoadjuvant chemotherapy, whereas all our hormone receptor positive patients received endocrine or combined endocrine with chemo therapy. Thus, it is expected that patients would have a better pathological response than with chemotherapy alone. Furthermore, associations between the hormone presence and pathological response to therapy is beyond the scope of the study.
- The Introduction section contains much unnecessary contents. Please summarize the Introduction and provide information around the main title of the study. The authors can organzie the Introduction section as follows:
- The importance of breast cancer in nowadays living
- The rate of locally advanced breast cancer (LABC)
- the importance of neoadjuvant chemotherapy in LABC
- grading systems of response to NAC.
- the importance of predicting the pathologic response to neoadjuvant chemotherapy (NAC)
- the lack of information regarding the the pathologic response to NAC
- Short definition of miRNAs.
- The current knowledge on the role of miRNAs in predicting response to chemotherapy (if no study on breast cancer, authors can use the other cancers in this section)
- the aim of the study
We have restructured the Introduction according to the reviewers indications
5. Introduction: "Treatment options for BC are represented by: surgery, radiotherapy, hormone or tar- 78 geted therapy and chemotherapy" Targeted therapy is not a common option in patients undergoing neoadjuvant chemotherapy.
We changed the paragraph: please see lines 58-65
Round 2
Reviewer 2 Report
Thanks for addressing the comments. The manuscript's quality is improved. However, comments no. 6, 10, 11, and 12 are not convincingly addressed in the revised manuscript. It is recommended to address all the comments.
Author Response
We would like to thank you for your time spent on reviewing our manuscript! Your comments and suggestions helped us to improve our paper! We are sorry for the error that occurred. We just noticed that the answer to suggestions 6-12 has not been loaded, We are deeply sorry You can find the responses below and some more adjustments. Please do not hesitate to contact us for other suggestions. Thank you!
6. Introduction: The rationale behind selecting the miRNA 375, 210, and let-7e in this study is not explained.
Thank you for this sugestion; We tried to clarify these aspects, in the Intoduction section, as follows
“However, recent treatment guidelines (15) encourage the administration of endocrine as well as targeted therapies concurrent with, or instead of NACT to increase tumours sensitivity to treatment. Thus, there is an increasing need to further explore the role of these miRNAs as biomarkers of NAT response. Based on existing literature, we have identified conflicting data regarding the prognostic role of several miRNAs. Of interest, miR-375-3p, miR-210-3p and let-7e have shown discrepancies regarding the clinical significance as prognostic biomarkers (16–20), being reported to have both increased and decreased expression associations with BC patients’ response to NAT. In this study, we aimed to assess the prognostic value of these highly controversial miRNAs, miR-375-3p, miR-210-3p and let-7e-5p, in breast cancer tissues by investigating their expression association with patients' pathological response to neoadjuvant therapy and clinico-pathological features.”
- Table 1. Please add the full term of TT in the footnote.
Thank you for this observation; we have clarified this term, as targeted therapy in the footnotes of Table 1.
8. Table 1 footnote: please correct the following typographical error:
"TNBC- Triple negativ" => negative
We have corrected as indicated
Table 2. Please choose a summarized heading (one line), and add the description to the footnote.
We have corrected as indicated
Table 2. Please add the reference for defining ER positivity >10% and PR positivity >20%. Based on my knowledge, ER and PR positivity are defined when >1% of the cancer cell are positive. This comment can be extended to KI-67. According to the literature, KI-67 >13% is considered high, not 20%.
According to the latest guidelines in the field, they are considered tumors with positive ER/PR at a value of >1%, but due to the clinical-pathological characteristics and outcome differences observed between patients with ER 1-9% and >10% and PR <20% and >20% and the difference in response to therapy, clinicians frequently use this value in the choice of therapy due to the cost-effective balance ( please see reference 21,22,23). Regarding ki-67 there is not a clear consensus for it’s cutoff, but at St Gallen International Expert Consensus, the majority voted for a threshold of 20 (please see reference 24. In our institute, at the time of data collection and processing, the guidelines that were followed had the aforementioned values ​​that guided the choice of therapy.
Discussion: Please add more details regarding the mechanistic pathways involved in the cancer biology. In case of lack of data, the authors can extrapolate the condition from other malignancies.
We suppose the reviewer is referring to the role of miRNAs in cancer biology. However, this aspect is a complex one and could be addressed specifically on molecular mechanisms of interest. Moreover, if we consider the multiple roles of miRNAs in modulating many targets simultaneously, is difficult to present details of mechanistic pathways involved in cancer biology. However, we pointed out in discussion section the main targets of miRNAs of interest, related to several specific mechanisms.
Discussion: In addition, it is recommended the authors add their own opinions for the machisms of action of the evaluated miRs in cancer biology. For example, recent literature have shown that mitochondria is at the center of cancer biology, in order that enhanced mitochondria biology can reduce cancer differentiation and improves treatment resistance ( https://www.preprints.org/manuscript/202201.0171/v3. and https://pubmed.ncbi.nlm.nih.gov/34890822/). On the other hand, it has been shown that miR-375 has an inhibitory effect of mitochondria biogenesis (https://pubmed.ncbi.nlm.nih.gov/22504094/). Therefore, this interaction can justify how miR-375 overexpression is associated with lower grade breast cancers and also high hormone receptor expression. Utmost, this interaction can justify how miR-375 can improve the response to neaodjuvant chemotherapy (https://www.preprints.org/manuscript/202201.0171/v3).
Thank you for this suggestion, we improve the Discution section with a paragraph, as follows:
“MiR-375 is a known tumor suppressor. In hepatocellular carcinoma (HCC), it inhibits the autophagy and tumor growth. Moreover, miR-375 promote the relaese of mitocondrial apoptotic proteins, reducing the vialbility of HCC cells in hypoxic conditions. In HCC, miR-375 was downregulated. Autophagy is an adaptive mechanism of the tumor cells that helps them to survive in the tumor microenvironment conditions by reducing the apoptosis and enhancing the elimination of the injured mithochondria (30). Although miR-375 could be related to treatment response by mitochondria reprogramming, to date there is no data presenting evidence about the role of mir-375 in inducing NAT response through mitochondria reprogramming in breast cancer.”
- Discussion: Please add the study strengths and limitations and provide suggestions for future works.
We have added a paragraph discussing these aspects at the end of the section.